# Clinicians' and Patients' Perceptions and Use of the Word "Cured" in Cancer Care: An Italian Survey

**Paolo Tralongo [1,*], Francesco Cappuccio [1], Stefania Gori [2], Vittorio Donato [3], Giordano Beretta [4], Ausilia Elia [1], Fabrizio Romano [1], Margherita Iacono [1], Antonino Carmelo Tralongo [1], Sebastiano Bordonaro [1], Annamaria Di Mari [1], Sebastiano Rametta Giuliano [1], Gabriella Buccafusca [1], Maria Carmela Careri [1] and Armando Santoro [5]**

[1] Medical Oncology Unit, Medical Oncology Department, Umberto I Hospital, RAO, 96011 Siracusa, Italy
[2] Medical Oncology Unit, IRCCS Sacro Cuore Don Calabria, Negrar di Valpolicella, 37024 Verona, Italy
[3] Radiotherapy Unit, San Camillo Forlanini Hospital, 00152 Rome, Italy
[4] Medical Oncology Unit, Santo Spirito Hospital, 65124 Pescara, Italy
[5] Department of Biomedical Sciences, Humanitas University, IRCCS Humanitas Research Hospital and Humanitas Cancer Center, 20089 Milan, Italy
*  Correspondence: paolo.tralongo@asp.sr.it

**Abstract:** Background. The words "hope" and "cure" were used in a greater number of articles and sentences in narrative and editorial papers than in primary research. Despite concomitant improvements in cancer outcomes, the related reluctance to use these terms in more scientifically oriented original reports may reflect a bias worthy of future exploration. This study aims to survey a group of physicians and cancer patients regarding their perception and use of the word cure. Materials and Method. An anonymous online and print survey was conducted to explore Italian clinicians' (the sample includes medical oncologists, radiotherapists, and oncological surgeons) and cancer patients' approach to the perception and use of the word "cure" in cancer care. The participants received an email informing them of the study's purpose and were invited to participate in the survey via a linked form. A portion, two-thirds, of questionnaires were also administered to patients in the traditional paper form. Results. The survey was completed by 224 clinicians (54 oncologists, 78 radiotherapists, and 92 cancer surgeons) and 249 patients. The results indicate a favourable attitude for patients in favour of a new language ("cured" vs. "complete remission") of the disease experience. Conclusions. The use of the word cured is substantially accepted and equally shared by doctors and patients. Its use can facilitate the elimination of metaphoric implications and toxic cancer-related connotations registered in all cultures that discourage patients from viewing cancer as a disease with varied outcomes, including cure.

**Keywords:** cured; survivorship; categorisation

## 1. Introduction

Fitzhugh Mullan, a physician, diagnosed with and treated for cancer himself, first described cancer survivorship as a concept; in 1986, he published a sentinel article titled "Season of Survival" describing his journey through what he described as acute, extended, and then permanent survivorship. Permanent survival, or "cured", was the last and final stage when individuals could be described as survivors [1]. He observed that for some cancer patients, there were late and long-term effects of cancer and/or its treatment [2].

Survivorship care models and guidelines for cancer patients are, nowadays, a growing health care and research priority in oncology.

In the past decade, diagnostic-therapeutic management has become more complex, and, more recently, the biological characterisation of the disease has assumed a relevant role. Its determination led to the proposal of drug treatments targeting specific receptors expressed by the cancer population. Due to this new clinical approach, accompanied by

an early diagnosis and the introduction of new effective drugs, the number of long-lived patients with a previous cancer diagnosis has increased considerably.

The overall trend of the estimated incidence in the current decade (+3.2% per annum) in Italy is comparable to that expected in the same period in the United States (+2.8% per annum), United Kingdom (+3.3%), and Switzerland (+2.5%) [3–5]. This trend underlines that this condition is not limited to Italy but concerns the whole world.

At the same time, using new language more suited to the current survivorship journey has been proposed [5,6].

The words "hope" and "cure" were used in a greater number of articles and sentences in narrative and editorial papers than in primary research, which highlights a relative reluctance to use these terms in more scientifically oriented original reports, despite concomitant improvements in cancer outcomes. This bias could be explored deeply in the future [7].

This study aims to survey a group of physicians (medical oncologists, radiotherapists, oncology surgeons) and cancer patients regarding their perception and use of the word "cure".

## 2. Materials and Methods

Between July and December 2021, an anonymous online and print survey was conducted to explore how Italian clinicians (the sample including medical oncologists, radiotherapists, and oncological surgeons) and cancer patients approach the perception and use of the word "cure" in cancer care. The participants received an email informing them of the study's purpose and were invited to take part in the survey via a linked form. A portion, two-thirds, of questionnaires were also only administered to patients in the traditional print form.

The questionnaire was developed taking into account the most important and current topics in the research area related to the language of cancer survivorship.

Physician participants are members of national scientific associations, such as AIOM (Associazione Italiana di Oncologia Medica), AIRO (Associazione Italiana Radioterapia e Oncologia clinica), SICO (Società Italiana di Chirurgia Oncologica), and patients association ROPI (Rete Oncologica Pazienti Italia) members. They were asked a series of eight questions regarding how commonly their patients are cured, how often they use the word "cure", in what circumstances they would tell a patient that they are cured, and about the timing of telling a patient that they are cured. The same questions were asked of cancer patients to investigate how confident they were with the concept and the use of the word "cure". An additional question was addressed to patients about insurance and financial issues.

The data reported in the tables have been standardised and summarised as percentages. Chi-squared tests, Fisher's exact test, and Cramer's Phi and V were used as statistical tools in order to compare responses between patients and the clinicians involved ($p < 0.005$).

The survey was completed by 224 clinicians (54 oncologists, 78 radiotherapists, and 92 cancer surgeons); a total of 249 patients were also involved in the study, of whom 100 completed the online form and 149 the print form. Table 1 reports the answers of all participants as a percentage.

For question number one on the possibility that all survivors live with the same conditions, a large percentage ranging between 96 and 97% is obtained.

For question number two, whether there can be cured patients among cancer patients, very high percentages are obtained: surgeons affirmed it at 98%, followed by radiotherapists with 97.4%, and by oncologists with 94.4%. Patients answered yes with a lower but still significant percentage, equal to 83.9%.

For question number three on what is meant by cured, patients are divided between "complete remission" and "equal risk of death of the non-cancer population". At the same time, the category of doctors expresses a concordant opinion, even if with an oscillation of about 10 points, respecting the definition of having the same risk of death as the population that has not been diagnosed with cancer by sex and age.

**Table 1.** Results.

| Questions | Patients | AIOM | AIRO | SICO |
|---|---|---|---|---|
| Are all patients alive after a previous diagnosis of cancer (survivors) in the same condition? | | | | |
| - Yes | 13.4% | 3.7% | 2.6% | 3% |
| - No | 86.6% ($p < 0.001$) | 96.3% ($p < 0.001$) | 97.4% ($p < 0.001$) | 97% ($p < 0.001$) |
| Are there any patients among them who could be defined as cured? | | | | |
| - Yes | 83.9% ($p < 0.001$) | 94.4% ($p < 0.001$) | 97.4% ($p < 0.001$) | 98% ($p < 0.001$) |
| - No | 16.1% | 5.6% | 2.6% | 2% |
| What do you mean by the term cured? | | | | |
| - patient with complete remission of disease | 35.7% | 27.8% | 15.4% | 17.5% |
| - patient with a low risk of recurrence | 14.9% | 2.2% | 9% | 4.9% |
| - patient with full recovery of health status | 18.5% | | | |
| - patient with a risk of death from cancer no greater than that of the general non-oncological population, of the same sex and equal age | 30.1% ($p < 0.001$) | 63% ($p < 0.001$) | 70.5% ($p < 0.001$) | 72.8% ($p < 0.001$) |
| - patient with 10-year disease remission | 0.8% | 3.5% | 3% | 3.8% |
| - patient with disease remission and low risk of recurrence | 0 | 3.5% | 2.1% | 1% |
| - (for patients only) cancer patients never cure | 0 | 0 | 0 | 0 |
| Are there any parameters for a cure for cancer? | | | | |
| - Yes | 53% ($p < 0.001$) | 74.1% ($p < 0.001$) | 67.9% ($p < 0.001$) | 80% ($p < 0.001$) |
| - No | 7.6% | 18.5% | 10.3% | 5% |
| - I don't know | 39.4% | 7.4% | 21.8% | 15% |
| If we consider the time variable, after how many years could the patient be considered as cured? | | | | |
| - 2 years | 0 | 0 | 0 | 0 |
| - 5 years | 19.3% | 20.4% | 7.7% | 10& |
| - 10 years | 8.4% | 11.1% | 11.5% | 15% |
| - 15 years | 2% | 0 | 0 | 5% |
| - variable according to the type of tumour | 70.3% ($p < 0.001$) | 68.5% ($p < 0.001$) | 80.8% ($p < 0.001$) | 70% ($p < 0.001$) |

**Table 1.** *Cont.*

| Questions | Patients | AIOM | AIRO | SICO |
|---|---|---|---|---|
| **If it is decided to apply the term cured, it should be used** | | | | |
| - during patient communication | 18.9% | 24.1% | 19.2% | 14% |
| - during scientific communication | 4.8% | 7.4% | 0 | 1% |
| - in both conditions | 76.3% ($p < 0.001$) | 68.5% ($p < 0.001$) | 80.8% ($p < 0.001$) | 85% ($p < 0.001$) |
| **How beneficial would its use be for the patient?** | | | | |
| - not at all | 3.2% | 1.9% | 0 | 1% |
| - a little | 14.5% | 11.1% | 7.7% | 9% |
| - a lot | 82.3% ($p < 0.001$) | 87% ($p < 0.001$) | 92.3% ($p < 0.001$) | 90% ($p < 0.001$) |
| **If considered beneficial, which aspect would the utility concern?** | | | | |
| - psychological discomfort | 26.1% | 14.8% | 11.5% | 10% |
| - social concerns | 3.4% | 2.6% | 0 | 0 |
| - better adherence to tertiary prevention interventions | 5% | 3% | 1.3% | 2% |
| - all of the above | 65.5% ($p < 0.001$) | 79.6% ($p < 0.001$) | 87.2% ($p < 0.001$) | 88% ($p < 0.001$) |
| **(for patients only) Have you had difficulty obtaining bank loans or life insurance?** | | | | |
| - Yes | 19.3% | | | |
| - No | 80.7% | | | |

For question number four, whether there are benchmarks for the definition of "cured", only 53% of patients answered yes compared to 39.4% who said they do not know, and 7.6% who said no, while among doctors, surgeons answered yes to 80% vs. 5% for no and 15% for I don't know, oncologists answered yes to 74% vs. 18.5% no and 7.4% I don't know, and radiotherapists answered yes to 67, 9% vs. 10.3% no and 21.8% don't know, with a discrepancy of about 13% points on yes. Regarding the I don't know, 39.4% are patients, 15% are surgeons, and 21.8% are radiotherapists.

Question five is about the time variable: the opinion is uneven since patients agree with percentages that differ slightly with oncologists and surgeons, around 70%, on the idea that the variable should be recalibrated concerning the type of cancer. In comparison, 80.8% of radiotherapists believe this. For patients and oncologists, about 20%, therefore, about a fifth of the sample per category, consider the 5-year parameter to be valid, which is of interest to only 10% of surgeons and 7.7% of radiotherapists.

On item number six, which induces a reflection on the context in which to use the word "healed" during a communication, there is broad agreement: scores of less than 10% emerge on exclusive use among professionals, that is, during scientific communications, with a predominance of the percentages in favour of both cases: both in communication between colleagues in scientific contexts and communications to patients, higher than 76% for patients and up to 85% for surgeons.

Question number seven induces reflection on the benefits for patients of using the term "cured". All respondents agree on the highly advantageous aspect that this use can have: oncologists at 87%, surgeons at 90%, and radiotherapists express the highest percentage equal to 92.3%, while patients, although settling on percentages higher than 80%, express themselves with a 14.5% towards the "little advantage".

Finally, item number eight invites us to consider the specific areas in which to find the possible benefits: the advantages would seem to be reflected in all the aspects considered, psychological, social, and adherence to subsequent preventive interventions, for all participants. Surgeons and radiotherapists agreed with percentages higher than 87%, while for a quarter of patients the advantages would be only with respect to psychological distress.

Figures 1–4 report a detailed description of the answers to the questionnaire items.

## In your opinion, all patients alive after a previous diagnosis of cancer (survivors) are in the same condition?

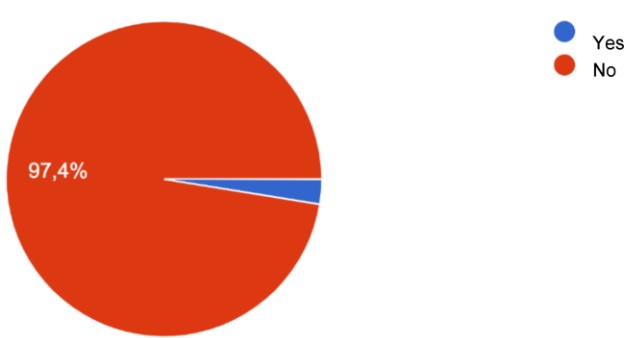

## Are there any patients among them who could be defined as cured?

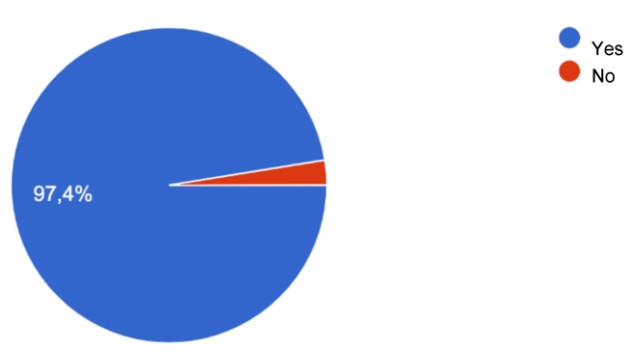

**Figure 1.** *Cont*.

## What do you mean by the term cured?

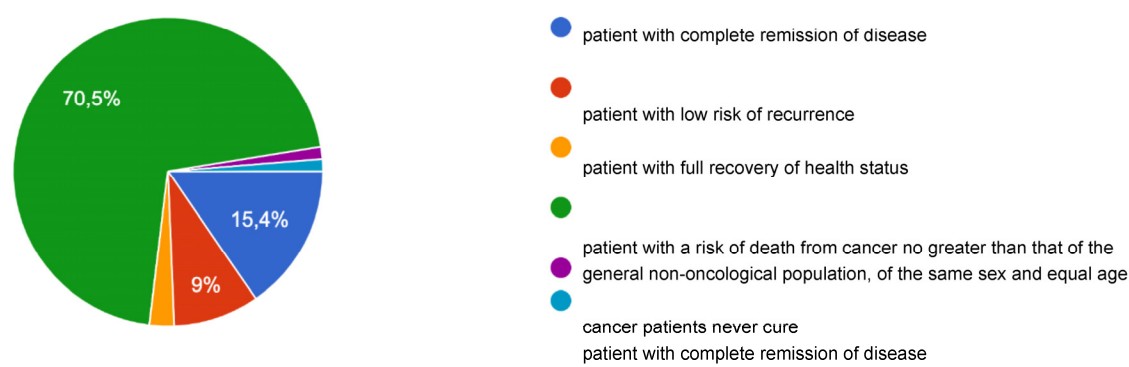

## Are there any parameters of cure from cancer?

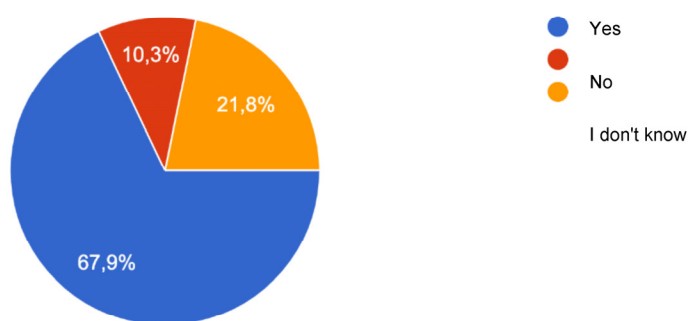

## If we consider the time variable, after how many years could the patient be considered as cured?

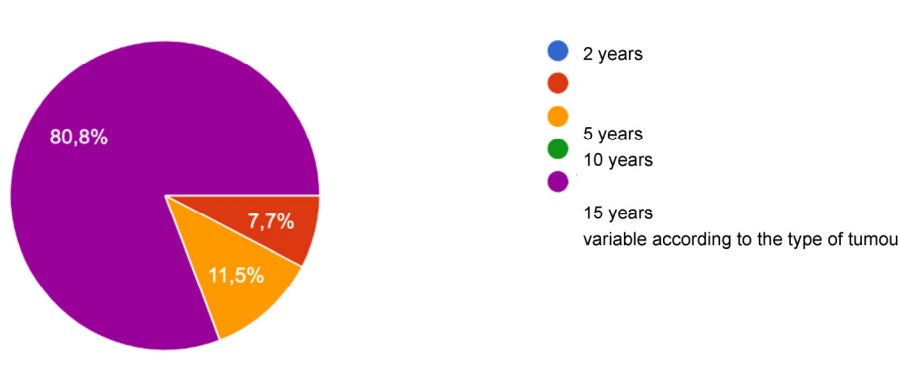

**Figure 1.** *Cont.*

## If it is decided to apply the term cured, it should be used

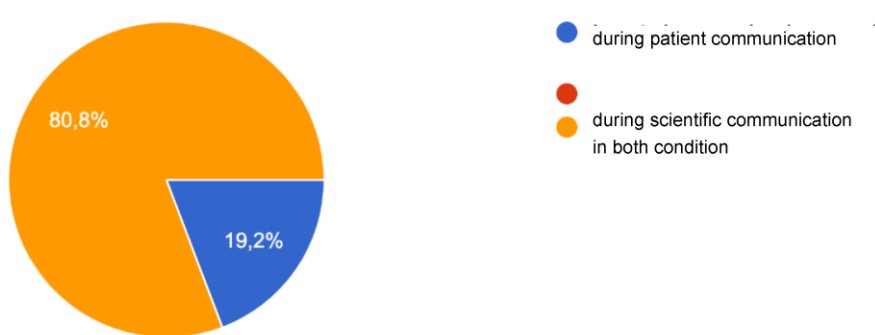

## How beneficial would be its use for the patient?

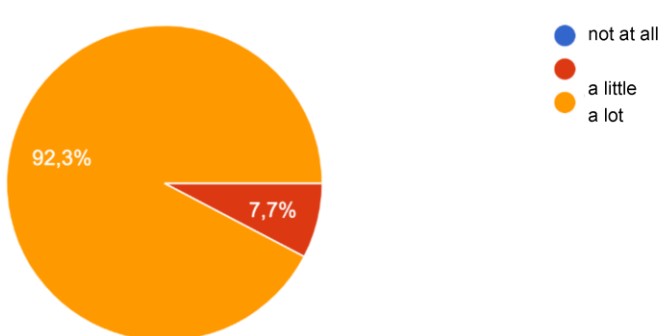

## If considered beneficial, which aspect would the utility concern?

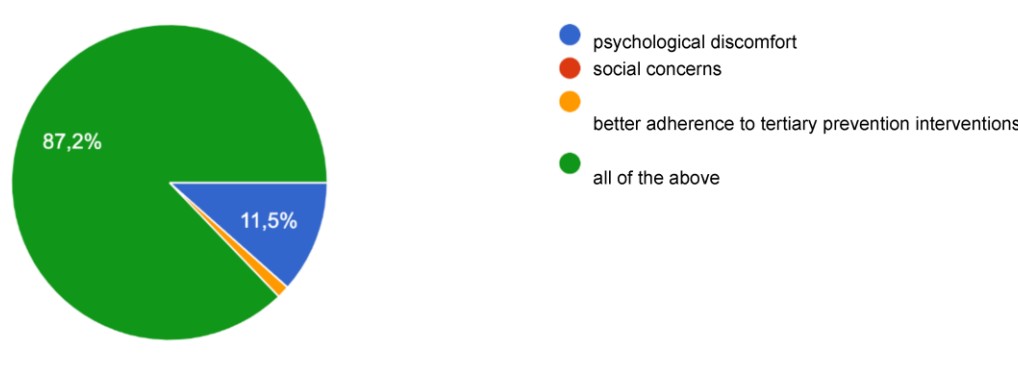

**Figure 1.** Radiotherapists.

## In your opinion, all patients alive after a previous diagnosis of cancer (survivors) are in the same condition?

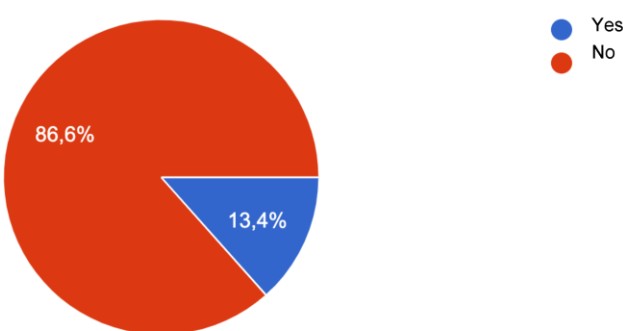

## Are there any patients among them who could be defined as cured?

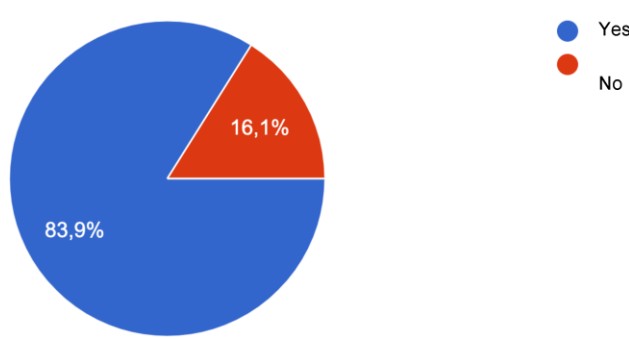

## What do you mean by the term cured?

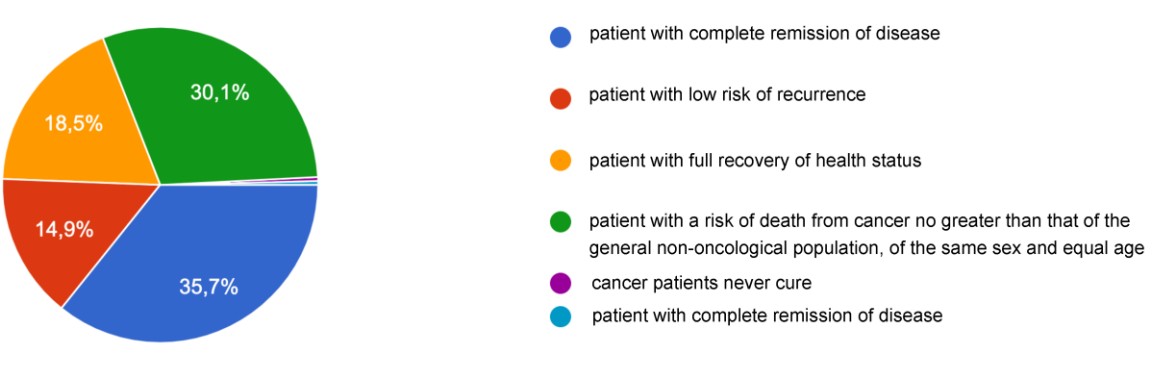

**Figure 2.** *Cont.*

## Are there any parameters of cure from cancer?

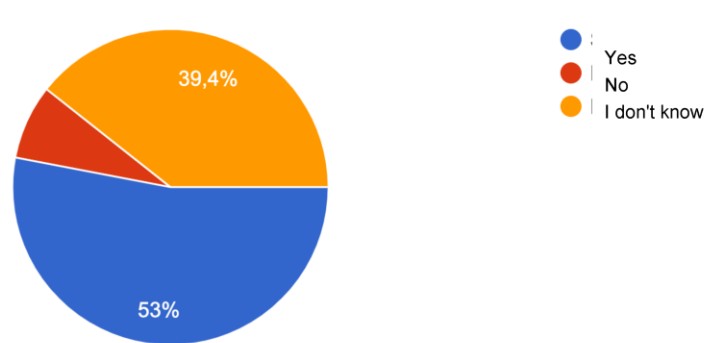

## If we consider the time variable, after how many years could the patient be considered as cured?

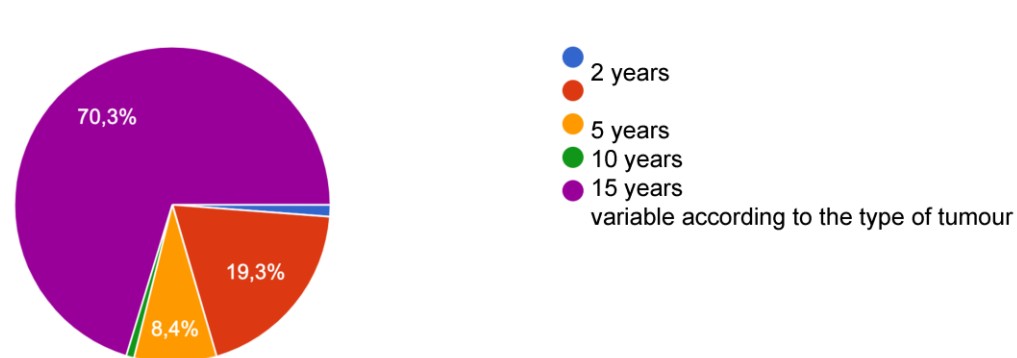

## If it is decided to apply the term cured, it should be used

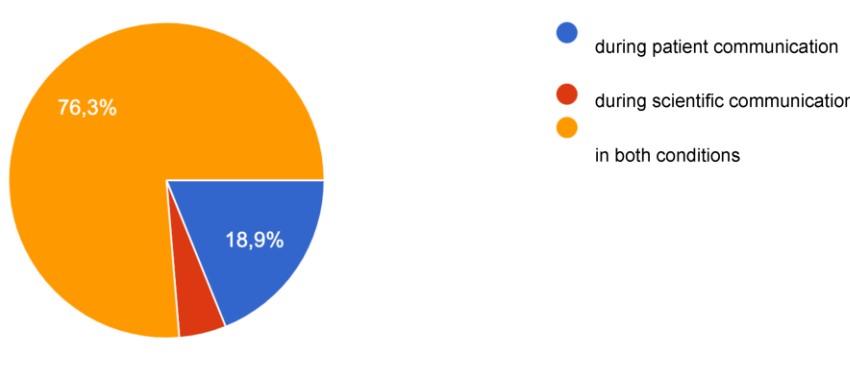

**Figure 2.** *Cont*.

## How beneficial would be its use for the patient?

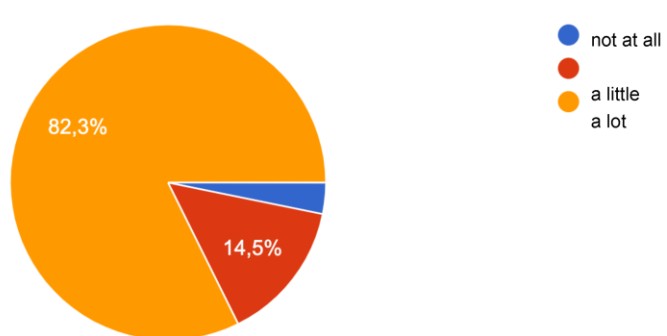

## If considered beneficial, which aspect would the utility concern?

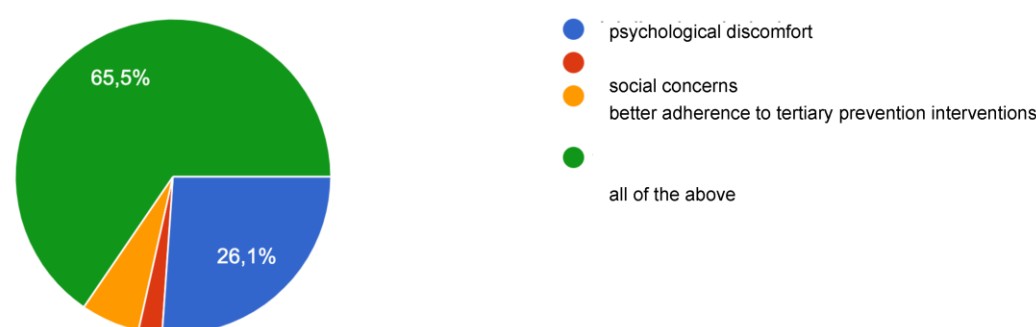

## Have you had difficulty getting bank mortgages or life insurance?

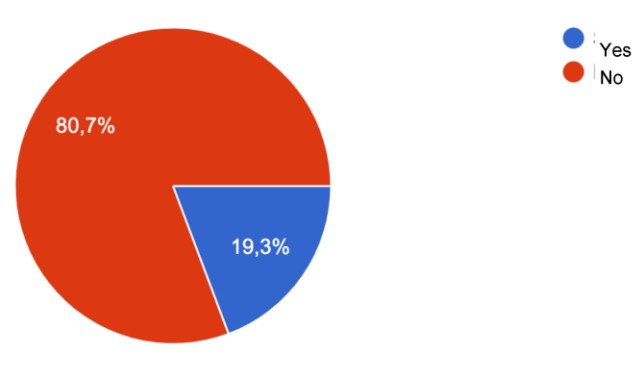

**Figure 2.** Patients.

# In your opinion, all patients alive after a previous diagnosis of cancer (survivors) are in the same condition?

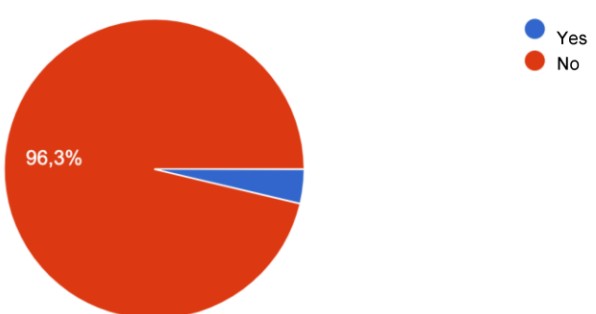

# Are there any patients among them who could be defined as cured?

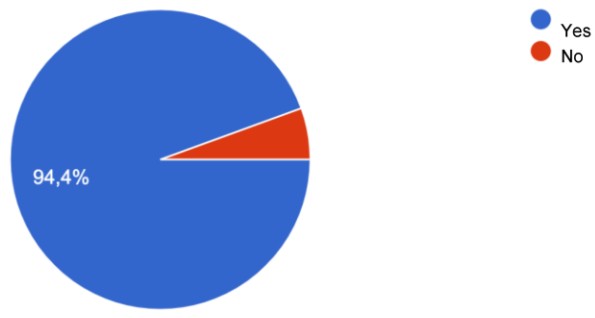

# What do you mean by the term cured?

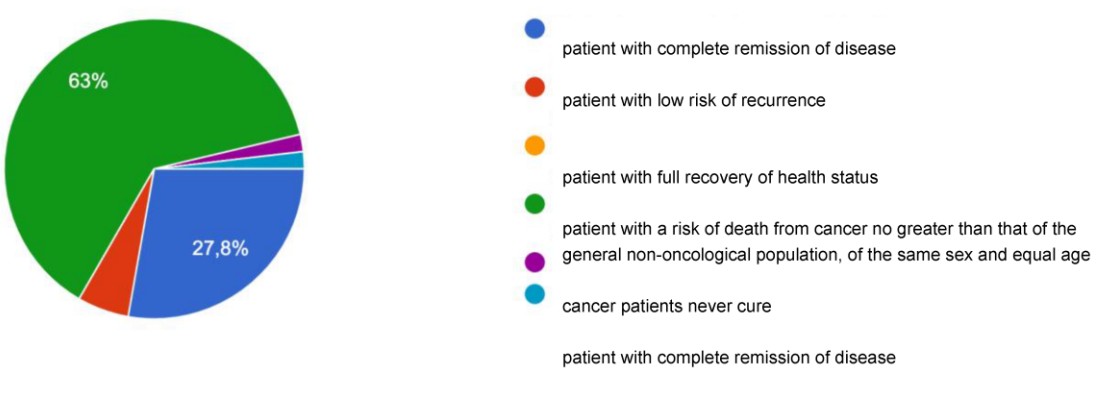

**Figure 3.** *Cont.*

# Are there any parameters of cure from cancer?

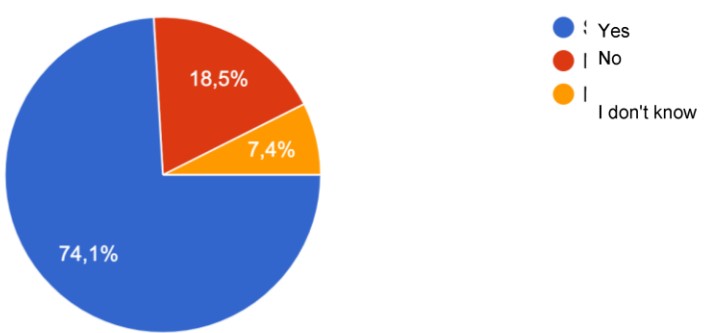

# If we consider the time variable, after how many years could the patient be considered as cured?

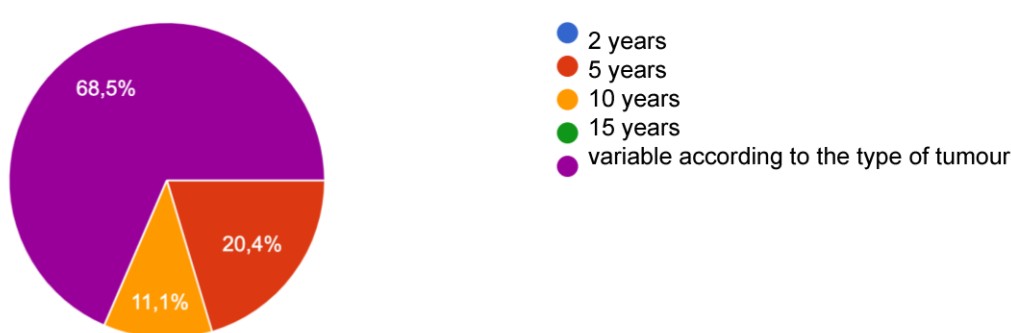

# If it is decided to apply the term cured, it should be used

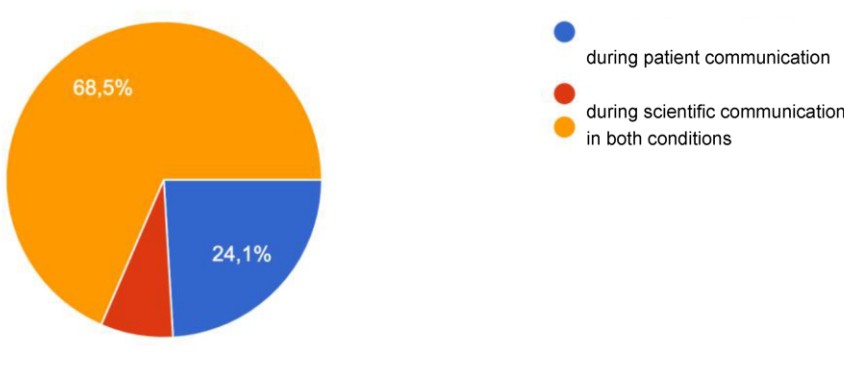

**Figure 3.** *Cont.*

## How beneficial would be its use for the patient?

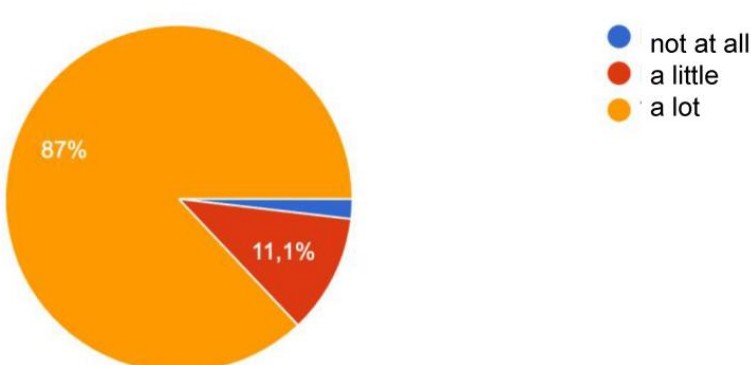

## If considered beneficial, which aspect would the utility concern?

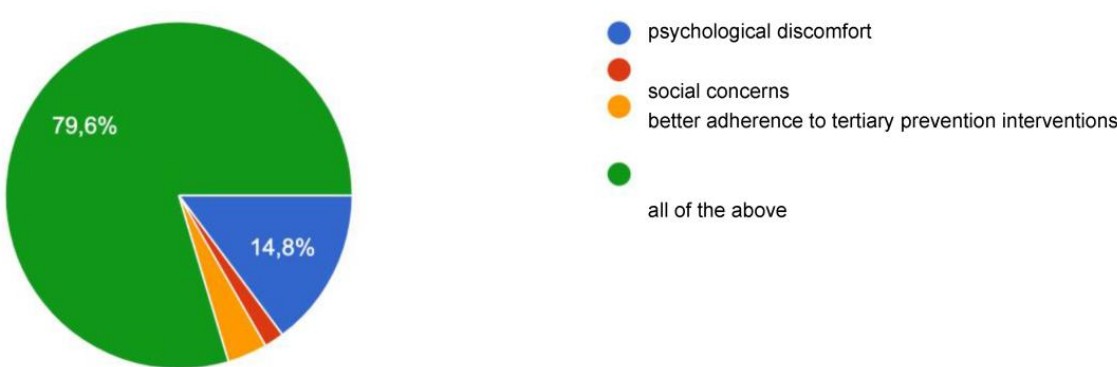

**Figure 3.** Oncologists.

The survey showed that the majority of participants (patients and clinicians) identify the concept of "cured" as part of the cancer disease pathway, cured meaning a patient with a risk of death from cancer disease no higher than that of the general population of the same sex and age after a number of years since diagnosis. Most participants answered that there are parameters that define a cure for cancer, and these can be defined not only by the variable of time since remission but also according to the characteristics of the tumour.

Responders agree on the need to use the term cured both during scientific communication and during communication with the patient as its use is considered advantageous for the patient from various points of view, from psychological discomfort to social issues and the consequent better adherence to tertiary prevention interventions.

The results highlight that there are essentially no differences and that both doctors and patients are oriented to using the term "cured" because they consider it appropriate in the clinical management of cancer disease.

Regarding social issues, 19.3% of patients report difficulties obtaining bank loans or life insurance.

In your opinion, all patients alive after a previous diagnosis of cancer (survivors) are in the same condition?

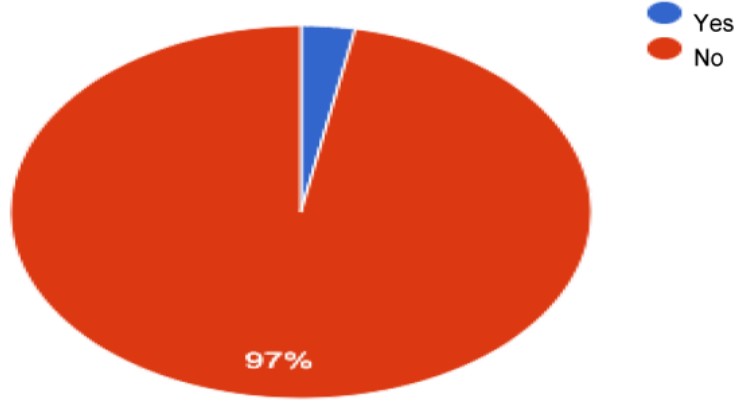

Are there any patients among them who could be defined as cured?

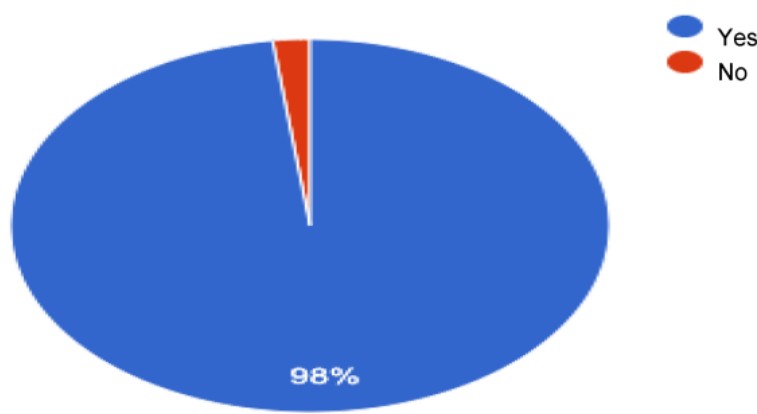

**Figure 4.** *Cont.*

## What do you mean by the term cured?

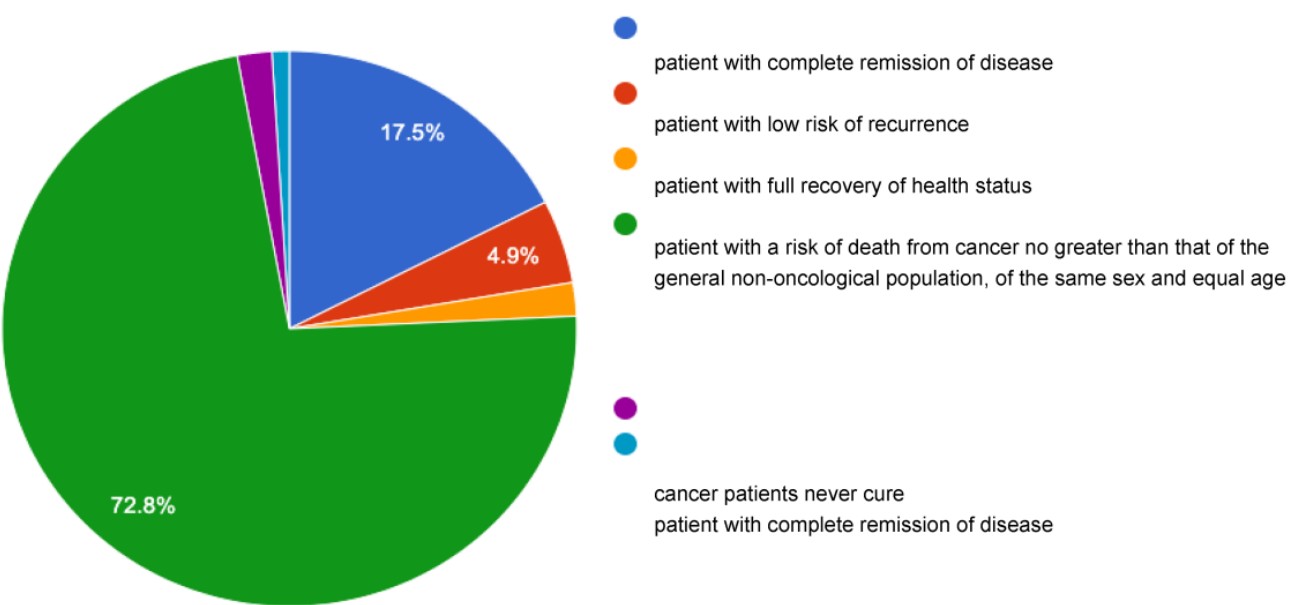

## Are there any parameters of cure from cancer?

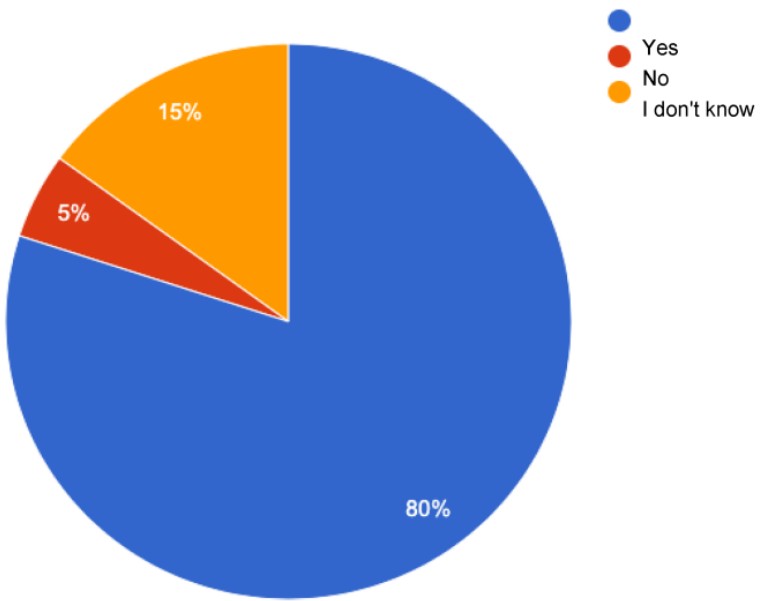

**Figure 4.** *Cont.*

If we consider the time variable, after how many years could the patient be considered as cured?

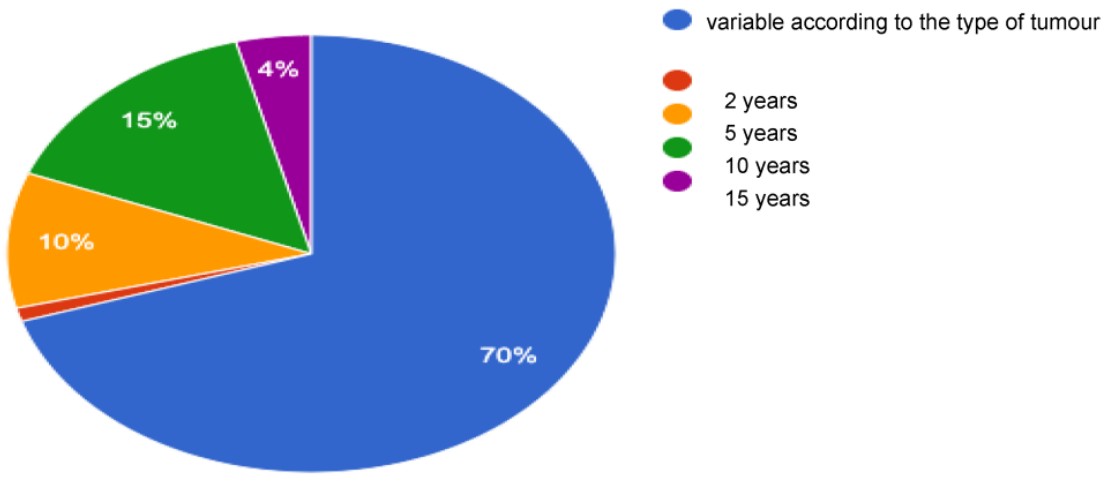

If it is decided to apply the term cured, it should be used

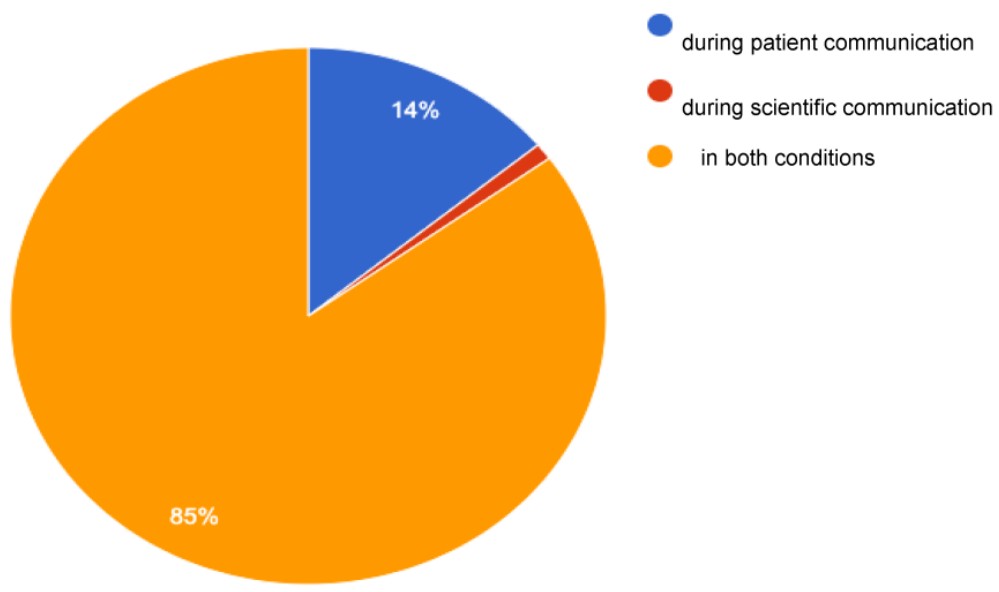

**Figure 4.** *Cont.*

## How beneficial would be its use for the patient?

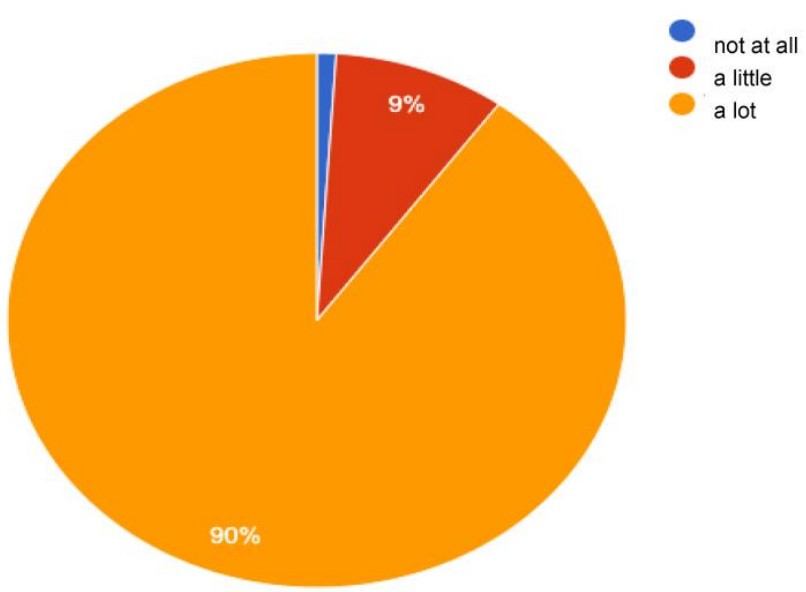

## If considered beneficial, which aspect would the utility concern?

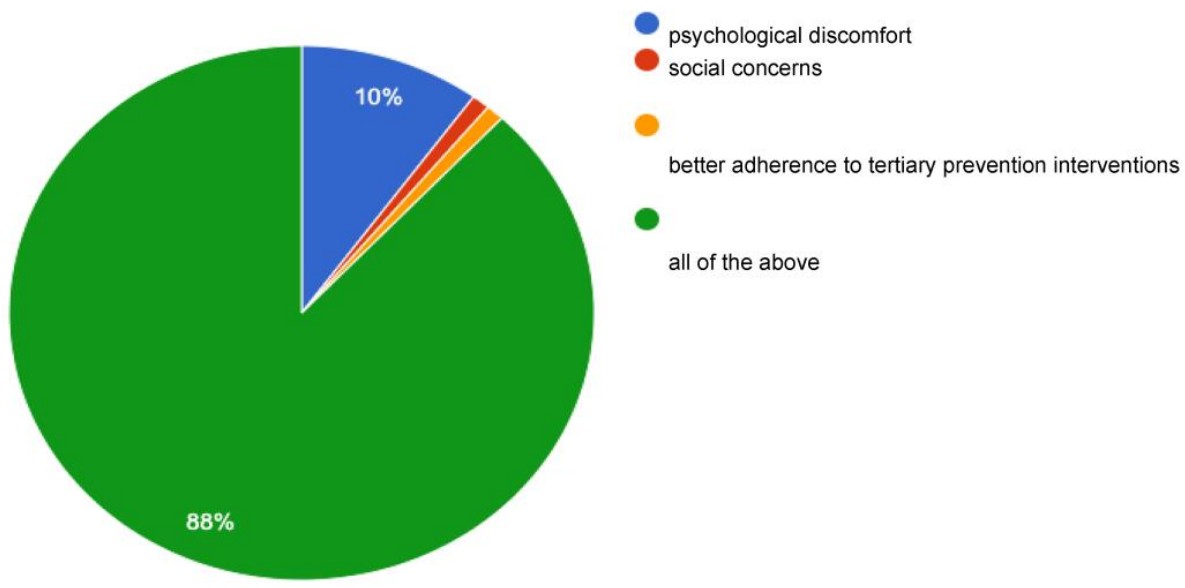

**Figure 4.** Surgeons.

## 3. Discussion

Long-term survivors often resume their usual life and jobs yet remain at risk for oncological, medical, rehabilitative, and psychosocial needs and issues [8,9]. In the last decade, the number of people with cancer who live longer than 5 years from diagnosis or end of acute treatment, along with those who live with cancer in a chronic state, has been increasing in industrialised countries. In Italy, they are about 3.6 million (3,609,135) people, representing 5.7% of the entire population, an increase of 37% compared to 10 years ago [10].

Consequently, survivorship care is nowadays a growing health care and research priority. It is now established that clinical and organisational categorisation of cancer survivors increases patients' physical and psychosocial well-being from a global sociocultural perspective [11].

The use of the term "cured" for some cancer patients is being debated in view of the increasing survival rates in some cancers [12] and the development of survivorship care as an essential component of oncology [13–15].

Clinicians are often reluctant to use the word "cured" in the clinical and communicative setting with patients, survivors, and their families. During communication with the patient, several terms are used for referring to those who are free of disease, including "in remission", "no evidence of disease", and "long-term survivor" instead of "cured". However, although patients prefer the term "cure", clinicians consider the use of the word "cure" as impossible in some settings [16] and thus are hesitant to tell them they are cured.

In the past years, some studies have evaluated the attitudes of oncology clinicians and patients about the use of the word cure in cancer care. A survey by the Dana-Farber Cancer Institute investigated the attitudes of 180 oncologists about the use of the word cure in cancer care [16]. Seventy-five per cent indicated that they were hesitant to use the word. Sixty-six per cent indicated that they would use the term when the risk of recurrence was either under 5% or essentially zero, and 20% reported that they never use the word.

Many respondents perceived that cancer survivors might be hesitant to be considered cured; 40% of clinicians indicated that their patients were hesitant to ask if they were or would be cured. Sixty per cent of respondents indicated that less than half of their patients ask if they are, or will be, cured. Some cited their own and their patients' hesitancy as related to the continued risk of relapse. There was concern among some respondents that some patients would react with superstition about indications that they were cured.

In our survey, patients agree with doctors regarding the cured definition with respect to the risk of death [17], even if about a third need to define themselves as cured just if they are in complete remission. It is as if the patient, touching his subjective emotionality, could not perceive himself cured except as "disease-free", seeing a negative evaluation in the hypothesis of "equal risk of the non-cancer population", which does not hold the hope of a qualitatively valid return to one's daily life. The "same risk" could correspond to the Sword of Damocles, which keeps the patient, even if healed, in the condition of psychological suspension of those who cannot "let their guard down". In this context, psychological rehabilitation paths that consolidate a more positive attitude for the subjective perception of being able to have resolved the risk of dying could be useful.

The opinion regarding the time variable is uneven since patients agree with percentages around 70% on the idea that the variable should be recalibrated with respect to the type of cancer. About 20% consider the 5-year parameter valid for patients and oncologists. The survey with item number four, relating to the possibility that there are reference parameters for the definition of "cured", observes a varied picture as shown in Figures 1–4: considering only the percentages of yes, it is observed that 53% of patients answered yes, while among doctors, 80% of surgeons, 74% of oncologists, and 67.9% of radiotherapists answered yes. Regarding the I don't know, 39.4% are patients, but as many as 15% are surgeons and 21.8% are radiotherapists, and it deserves to be supported by education. The question then arises of continuing education that allows homogeneity not only of the words to be adopted but also of the overall vision towards which to accompany patients and family members.

Item six further explores the theme of the context in which to use the word "cured" during communication: there would seem to be broad agreement (percentage scores below 10%) that it is not just an element to be used among professionals but both in communication between colleagues in scientific contexts and communications to patients. Oncologists highlight a noteworthy element: for about a quarter of their responses (24.1%), it should be used only in communications with patients and not in scientific communications.

This consideration calls for educational interventions on the usefulness of the cross-use of the word cured to eliminate the social discrimination to which these patients are still subjected.

Question number seven induces reflection on the benefits for patients of the use of the term "cured": all the interviewees agree on the highly advantageous aspect that this use can have, but it is the patients who, despite settling on percentages above 80%, show greater scepticism than doctors, with 14.5% towards the little advantage, reporting the theme of "being able to believe" in the value, not only literal, of the term to underline the need for a communicative intervention that transfers knowledge to patients.

With regard to the question given to patients only on the effects of the disease on economic and financial needs, about 20% confirm the need for legislative intervention that avoids this social discrimination [14,18,19].

Specifically, the advantages would seem to be reflected in all the aspects considered, psychological, social, and adherence to subsequent preventive interventions, for all participants; however, surgeons and radiotherapists agreed with percentages greater than 87%, while for a quarter of patients, the advantages would be only with respect to psychological distress.

Patients would differentiate the answers more on their individual personal experiences. Regarding the term "cured", everyone thinks slightly the same way and therefore they support its use in communication settings with the patient and his family and scientific communication, albeit with different shades. The word "cured" usually refers to complete clinical remission of cancer, regardless of the presence or absence of late sequelae of treatments [9]. In this survey, a cancer patient can be defined as "cured" only when his or her life expectancy is the same as the sex- and age-matched general population [17].

Regardless, achieving cured status does not mean that the patient is dropped from surveillance but rather selecting the interventions (tertiary prevention, promotion of lifestyles, starting rehabilitation interventions, . . . ) most suited to his condition to create, in the era of precision medicine, a tailored intervention.

Nowadays, the possibility of determining the time necessary to reach the cured condition is proposed. In this regard, few studies have categorised prevalent cancer patients according to the probability of being cured. The EUROCARE-5 study included information on cancer patients diagnosed in 29 European countries and 99 population-based cancer registries (CRs) [13]. It included 7.2 million adult (aged 15–74 years) cancer patients, with 15 years of registration during 1990–2007 and 18 years of follow-up as of 2008, collected by 42 CRs from 17 countries and 19% of the European population [20,21]. The study aimed to provide reliable population-based estimates of three indicators of long-term prognosis and cure of cancer patients in Europe by cancer type, sex, and age. They serve as 'real-world' information addressed to health professionals for evaluating treatment effectiveness, to oncologists for planning follow-up, and policymakers for allocating resources. According to estimates of CF ("cure fraction"—the proportion of cancer patients having the same mortality rates as those observed in the general population of the same sex and age) and TTC ("time to cure"—the number of years after cancer diagnosis when the excess mortality due to cancer becomes negligible), four major clusters of cancer types emerged. The first included testicular or thyroid cancer patients, for whom surveillance may be warranted only for the first 1–2 years since no relevant excess mortality would persist later. The cure was also reached by more than two-thirds of patients with melanoma, HL, cancer of the cervix uteri, colorectal, and endometrial cancers, for which a cure is achieved by approximately half of patients with TTC < 10 years, suggesting the need for medium-term

surveillance. A third cluster included patients with breast, bladder, and prostate cancers, since, consistently across studies, 50–70% of them were not expected to die because of their neoplasms, but a small risk of death persisted for at least 15 years.

The fourth major cluster included patients with liver, lung, and pancreatic cancers with a median of 4–6 months of survival. These cancers showed small CF changes during the observation period.

Categorisation, based on clinical, epidemiologic, and death risk-assessment data, is not antagonistic to existing inclusive definitions of cancer survivorship. Rather, it may complement them by allowing tailored survivorship care to be delivered effectively and sensitively to different individuals belonging to different categories by evaluating their actual disease and risk status, which is now increasingly possible [22–26]. On the other hand, from the functional point of view, there is a need to define cancer survivors' disease phases (acute, chronic, cured) for the practical purpose of designing and implementing appropriate models of care or follow-up guidelines [7,27,28]. When appropriate, we believe the word cured can be used in the clinical setting during communication with patients and their families but should always be accompanied by counselling about prevention, screening, and maintenance of good general health for all, as suggested in this survey differently from what was reported in a previous experience, where oncology clinicians reported that patients are hesitant to ask whether they are cured, while the clinicians are hesitant to tell patients they are cured [16].

The study has limitations in the number of subjects who participated in the survey. A subsequent evaluation of a larger population should be developed to confirm these results. In any case, the multidisciplinary comparison has provided interesting data.

Patients and clinicians are essentially of the same opinion: they consider that the time has come to use the term cured, in the condition in which this definition is applicable; this is probably because the most recent data on cancer patients' life expectancy helped reduce resistance to the use of the word "cured". After all, they now recognise in its use some advantages such as the promotion of tertiary prevention and cancer rehabilitation, reductions of social discrimination, and modification and personalisation of cancer surveillance by focusing it on the patient as well as on the cancer disease.

## 4. Conclusions

Although the actual number of cancer survivors that may appropriately be defined as cured is limited, we could now concretely apply this term to specific clinical situations and in organisational settings and policy making, with potential positive reverberations in the human and relational dimensions of the lives of patients.

Indeed, the correct interpretation of the word "cured" is important because it could facilitate the return of individual patients to their social and professional life as before cancer, reducing the risk of employers, insurance, and social discrimination and promoting personalised surveillance and health promotion interventions.

Their use and, in a broader perspective, the categorisation of cancer survivors requires a paradigm shift in the culture of cancer survivorship care, whereby we abandon a common, general approach to all cancer patients in favour of the application of epidemiologic knowledge and the developing risk assessment tools to tailor follow-up and recommendations to each cancer survivor. That is why patients who belong to different categories cannot be treated and observed similarly [15–17].

This study evaluates the use and perceptions of the word cure among academic and community doctors such as surgeons, oncologists, and radiotherapists and cancer survivors of different institutions in Italy.

The results indicate a favourable attitude, for all participants, in favour of a new language of the disease experience that, in turn, facilitates the elimination of metaphoric implications and toxic cancer-related connotations registered in all cultures that discourage patients from viewing cancer as a disease with varied outcomes, including cure [7,11,27–29]. However, it could be useful for the patients to start psychological

rehabilitation programs aimed at favouring a more positive view of them with respect to the life expectancy they can reach.

Finally, demystifying the word cancer with the use of word "cured" will require the concerted efforts of physicians, patients, families, and media to create a new culture in oncology and help to move survivorship care into personalised precision approaches that will improve, in turn, the medical and psychosocial care for each survivor and will help reduce the stigma of a disease that still persists in many cultures [16].

**Author Contributions:** Conceptualization, P.T.; Methodology, P.T.; Validation, G.B. (Giordano Beretta) and V.D.; Investigation, A.E.; Resources, F.R.; Data curation, F.C., M.I. and S.G.; Writing original draft, A.C.T. and P.T.; Writing and review & editing, S.B., A.D.M. and S.R.G.; Visualization, G.B. (Gabriella Buccafusca); Supervision, M.C.C.; Project administration, A.S. All authors have read and agreed to the published version of the manuscript.

**Funding:** This research received no external funding.

**Institutional Review Board Statement:** Not applicable.

**Informed Consent Statement:** Not applicable.

**Data Availability Statement:** Data supporting reported results can be request to the corresponding author.

**Conflicts of Interest:** The authors declare no conflict of interest.

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
