# Peer review of "Clinicians’ and Patients’ Perceptions and Use of the Word “Cured” in Cancer Care: An Italian Survey"

_curroncol, doi:10.3390/curroncol30020103_

Round 1
Reviewer 1 Report
The topic could be interesting for the readers, I have concerns related to statistical relevance of the study and related to the consequences (e.g. cured=not going to follow up checking) if the work is published. For these reasons I feel like requesting to involve a psychologist and/or an ethical committee.
Please find the revised manuscript in the attachment.

Author Response
In the Discussion section a note has been reported which explains that the treated patient does not suspend the follow-up but a personalized one is planned for him.
The bibliography has been revised.

Reviewer 2 Report
Very interesting and informative article that is clearly presented.
Significant proof-reading is advised to correct numerous typos and ungrammatical sentences.
Figs 2, 3 and 4 are illegible in current format. I would suggest verticalizing them so they occupy a bigger space and enlarge font size.
Both the discussion and the conclusion sections are a bit too long for the actual data load of the study. Some parts within the early-mid Discussion sound like a repetition of the described results in the previous section. Later, the Discussion ends in a tone again picked up in the Conclusions, which could be written more concisely.
Author Response
We have reviewed the results and discussion sections, omitting the redundancies.
The conclusions have been reported more concisely, as requested.
The figures have been reported vertically.

Round 2
Reviewer 1 Report
the suggested improvements were little considered, even figure were not improved....